# Q-LEARNING DECISION TRANSFORMER: LEVERAGING DYNAMIC PROGRAMMING FOR CONDITIONAL SEQUENCE MODELLING IN OFFLINE RL

## ABSTRACT

Recent works have shown that tackling offline reinforcement learning (RL) with a conditional policy produces promising results. The Decision Transformer (DT) combines the conditional policy approach and a transformer architecture, showing competitive performance against several benchmarks. However, DT lacks stitching ability – one of the critical abilities for offline RL to learn the optimal policy from sub-optimal trajectories. This issue becomes particularly significant when the offline dataset only contains sub-optimal trajectories. On the other hand, the conventional RL approaches based on Dynamic Programming (such as Q-learning) do not have the same limitation; however, they suffer from unstable learning behaviours, especially when they rely on function approximation in an off-policy learning setting. In this paper, we propose the Q-learning Decision Transformer (QDT) to address the shortcomings of DT by leveraging the benefits of Dynamic Programming (Q-learning). It utilises the Dynamic Programming results to relabel the return-to-go in the training data to then train the DT with the relabelled data. Our approach efficiently exploits the benefits of these two approaches and compensates for each other's shortcomings to achieve better performance. We empirically show these in both simple toy environments and the more complex D4RL benchmark, showing competitive performance gains.

## 1 INTRODUCTION

The transformer architecture employs a self-attention mechanism to extract relevant information from high-dimensional data. It achieves state-of-the-art performance in a variety of applications, including natural language processing (NLP) (Vaswani et al., 2017; Radford et al., 2018; Devlin et al., 2018) or computer vision (Ramesh et al., 2021). Its translation to the RL domain, the Decision transformer (DT) (Chen et al., 2021), successfully applies the transformer architecture to offline reinforcement learning tasks with good performance when shifting their focus on the sequential modelling. It employs a goal conditioned policy which converts offline RL into a supervised learning task, and it avoids the stability issues related to bootstrapping for the long term credit assignment (Srivastava et al., 2019; Kumar et al., 2019b; Ghosh et al., 2019). More specifically, DT considers a sum of the future rewards – return-to-go (RTG), as the goal and learns a policy conditioned on the RTG and the state. It is categorised as a *reward conditioning* approach.

Although DT shows very competitive performance in the offline reinforcement learning (RL) tasks, it fails to achieve one of the desired properties of offline RL agents, stitching. This property is an ability to combine parts of sub-optimal trajectories and produce an optimal one (Fu et al., 2020). Below, we show a simple example of how DT (*reward conditioning* approaches) would fail to find the optimal path.

To demonstrate the limitation of the reward conditioning approaches (DT), consider a task to find the shortest path from the left-most state to the rightmost state without going down to the fail state in Fig. 1. We set the reward as $-1$ at every time step and $-10$ for the action going down to the fail state. The training data covers the optimal path, but none of the training data trajectories has the entire optimal path. The agent needs to combine these two trajectories and come up with the optimal path. The reward conditioning approach essentially finds a trajectory from the training data

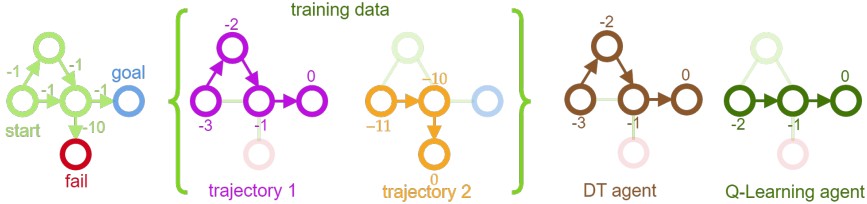

Figure 1: A simple example demonstrates the decision transformer approach's issue (lack of *stitching* ability) – fails to find the shortest path to the goal. In contrast, Q-learning finds the shortest path. The numbers on the arrows are rewards on the path and the numbers on the states are RTGs.

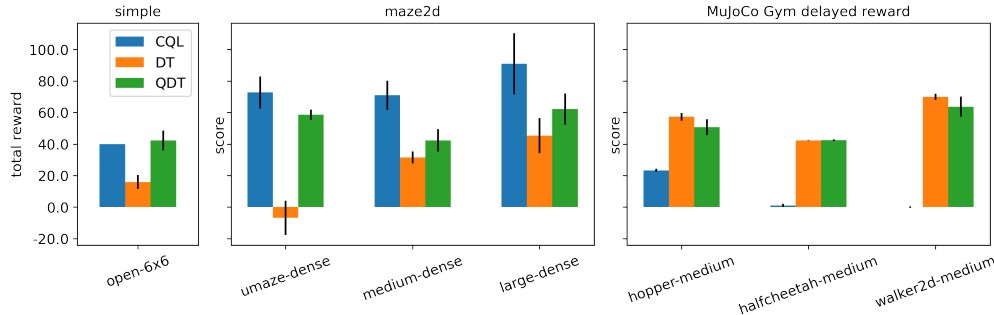

Figure 2: Evaluation results for conservative Q-learning (CQL), Decision Transformer (DT) and Q-learning Decision Transformer (QDT). The left two plots (simple and maze2d environments) show that the DT does not perform as it fails to stitch trajectories, and the right plot shows that CQL fails to learn from a sparse reward scenario (delayed reward). In contrast, QDT achieves consistently good results across all the environments.

that gives the ideal reward and takes the same action as the trajectory. In this simple example, trajectory 2 has a meagre reward. Hence, it always follows the path of trajectory 1 despite trajectory 2 giving the optimal path for the first action.

In contrast to the reward conditioning approaches (DT), Q-learning[1] does not suffer from the issue and finds the optimal path quickly in this simple example. Q-learning takes each time step separately and propagates the best future rewards backwards. Hence it can learn from the first optimal action from trajectory 2. However, Q-learning has some issues on a long time horizon and sparse reward scenarios. It attempts propagating the value function backwards to its initial state, often struggling to learn across long time horizons and sparse reward tasks. This is especially true when Q-learning uses function approximation in an off-policy setting as discussed in Section 11.3 in (Sutton & Barto, 1998).

Here, we devise a method to address the issues above by leveraging Q-learning to improve DT. Our approach differs from other offline RL algorithms that often propose a new single architecture of the agent and achieves better performance. We propose a framework that improves the quality of the offline dataset and obtains better performance from the existing offline RL algorithms. Our approach exploits the Q-learning estimates to relabel the RTG in the training data for the DT agent. The motivation for this comes from the fact that Q-learning learns RTG value for the optimal policy. This suggests that relabelling the RTG in the training data with the learned RTG should resolve the DT stitching issue. However, Q-learning also struggles in situations where the states require a large time step backward propagation. In these cases, we argue that DT will help as it estimates the sequence of states and actions without backward propagation. Our proposal (QDT) exploits the strengths of each of the two different approaches to compensate for other's weaknesses and achieve a more robust performance. Our main evaluation results are summarised in Fig. 2. The left two

---

[1]In this paper, we will use the *Q-learning* and *Dynamic Programming* interchangeably to indicate any RL algorithm relying on the Bellman-backup operation.

plots (simple and maze2d environments) show that DT does not perform well as it fails to stitch trajectories, while the right plot illustrates that CQL (Q-learning algorithm for offline reinforcement learning) fails to learn in a sparse reward scenario (delayed reward). These results indicate that neither of these approaches works well for all environments, and we might have abysmal results by selecting the wrong type of algorithms. In contrast, QDT performs consistently well across all environments and shows robustness against different environments. Through our evaluations, we also find that some of the evaluation results in the prior works may not be directly comparable, and it causes some contradicting conclusions. We touch on the issue in Section 6.

## 2 PRELIMINARIES

**Offline Reinforcement Learning.** The goal of RL is to learn a policy that maximises the expected sum of rewards in a Markov decision process (MDP), which is a four-tuple $(S, A, p, r)$ where $S$ is a set of states, $A$ is a set of actions, $p$ is the state transition probabilities, and $r$ is a reward function.

In the online or on-policy RL settings, an agent has access to the target environment and collects a new set of trajectories every time it updates its policy. The trajectory consists of $\{s_t, a_t, r_t\}_{t=0}^{T}$ where $s_t$, $a_t$ and $r_t$ are the state, action and reward at time $t$ respectively, and $T$ is the episode time horizon.

In off-policy RL case, the agent also has access to the environment to collect trajectories, but it can update its policy with the trajectories collected with other policies. Hence, it improves its sample efficiency as it can still make use of past trajectories.

Offline RL goes one step further than off-policy RL. It learns its policy purely from a static dataset that is previously collected with an unknown behaviour policy (or policies). This paradigm can be precious in case of the interaction with the environment being expensive or high risk (e.g., safety critical applications).

**Decision Transformers.** DT architecture (Chen et al., 2021) casts the RL problem as conditional sequence modelling. Unlike the majority of prior RL approaches that estimates value functions or compute policy gradients, DT outputs desired future actions from the target sum of future rewards RTGs, past states and actions.

$$\tau = (R_{t-K+1}, s_{t-K+1}, a_{t-K+1}, \cdots, R_{t-1}, s_{t-1}, a_{t-1}, R_t, s_t) . \tag{1}$$

Equation 1 shows the input of a DT, where $K$ is the context length, $R$ is RTGs ($R_t = \sum_{t'=t}^{T} r_{t'}$), $s$ is states and $a$ is actions. Then DT outputs the next action ($a_t$).

DT employs Transformer architecture (Vaswani et al., 2017), which consists of stacked self-attention layers with residual connections. It has been shown that the Transformer architecture successfully relates scattered information in long input sequences and produces accurate outputs (Vaswani et al., 2017; Radford et al., 2018; Devlin et al., 2018; Ramesh et al., 2021).

**Conservative Q learning.** In this work, we use the conservative Q learning (CQL) framework (Kumar et al., 2020) for the Q-learning algorithm. CQL is an offline RL framework that learns Q-functions that are lower-bounds of the true values. It augments the standard Bellman error objective with a regulariser which reduces the value function for the out-of-distribution state-action pair while maintaining ones for state-action pairs in the distribution of the training dataset. In practice, it uses the following iterative update equation to learn the Q-function under a learning policy $\mu(a|s)$.

$$\hat{Q}^{k+1} \leftarrow arg \min_{Q} \alpha \left( \mathbb{E}_{s\sim\mathcal{D}, a\sim\mu(a|s)}[Q(s, a)] - \mathbb{E}_{s, a\sim\mathcal{D}}[Q(s, a)] \right)$$
$$+ \frac{1}{2} \mathbb{E}_{\substack{s, a, s' \sim \mathcal{D} \\ a' \sim \mu(a'|s')}} \left[ \left( r(s, a) + \gamma\hat{Q}^k(s', a') - Q(s, a) \right)^2 \right], \tag{2}$$

where $\mathcal{D}$ is the training dataset and $\gamma$ is a discount factor. Kumar et al. (2020) showed that while the resulting Q-function, $\hat{Q}^{\mu} := \lim_{k\to\infty} \hat{Q}^k$ may not be a point-wise lower-bound, it is a lower bound of $V(s)$, i.e. $\mathbb{E}_{\mu(a|s)}[\hat{Q}^{\mu}(s, a)] \leq V^{\mu}(s)$.

## 3 METHOD

We propose a method that leverages Dynamic Programming approach (Q-learning) to compensate for the shortcomings of the reward conditioning approach (DT) and build a robust algorithm for the offline RL setting. Our proposal consists of three steps. First, the value function is learned with Q-learning. Second, the offline RL dataset is refined by relabelling the RTG values with the result of Q-learning. Finally, the DT is trained with the relabelled dataset. The first and third steps do not require any modifications of the existing algorithms.

The reward conditioning approach (DT) takes an entire trajectory sequence and conditions on it using the sum of the rewards for that given sequence. Such an approach struggles on tasks requiring *stitching* (Fu et al., 2020) – the ability to learn an optimal policy from sub-optimal trajectories by combining them. In contrast, the Q-learning alternative propagates the value function backwards for each time step separately with the Bellman backup, and pools the information for each state across trajectories. It therefore does not have the same issue. Our proposal tackles the *stitching* issue of the reward conditioning approach by relabelling the RTG values with the learned Q-functions. With the relabelled dataset, the reward conditioning approach (DT) can then utilize optimal sub-trajectories from their respective sub-optimal trajectories.

We now discuss how to relabel the RTGs values with the learned Q-functions. Replacing all of the RTGs values with Q-functions is not adequate because not all the learned Q-functions are accurate, especially in a long time horizon and sparse reward case. Ideally, we would like to replace the RTGs values where the learned Q-functions are accurate. In this work, we employ the CQL framework for the offline Q-learning algorithm, which learns the lower bound of the value function. We replace the RTGs values when the RTG in the trajectory is lower than the lower bound. With this approach, our method substitutes the RTGs values where the learned value function is indeed accurate (or closer to the true values). We also replace all RTG values prior to the replaced RTG along with the trajectory by using reward recursion ($R_{t-1} = r_{t-1} + R_t$). This propagates the replaced RTG values to all the time steps prior to the replaced point. To apply this idea, we initialise the last state RTG to zero ($R_T = 0$), and then we start the following process from the end of the trajectory to the initial state backwards in time. First, the state value is computed for the current state with the learned value function $\hat{V}(s_t) = \mathbb{E}_{a \sim \pi(a|s_t)}[\hat{Q}(s_t, a)]$, where the $\pi$ is the learned policy. Next, the value function is compared ($\hat{V}(s_t)$) against the RTG value for the current state ($R_t$). If the value function is greater than that of the RTG, the RTG for the previous time step is set from ($R_{t-1}$) to $r_{t-1} + \hat{V}(s_t)$, otherwise it is set to $r_{t-1} + R_t$. We repeat this process until the initial state is reached. This process is summarised in Algorithm 1.

The above relabelling process might introduce inconsistencies between the reward and RTG within the DT input sequence (Eq. 1). The RTG value is sum of the future rewards, hence it must always be $R_t = r_t + R_{t+1}$. However, the relabelling process might break this relationship. To maintain this consistency within the input sequence of DT, we regenerate the RTG for the input sequence ($\{\hat{R}_{t-K+1}, \cdots, \hat{R}_{t-1}, \hat{R}_t\}$) by copying the last RTG ($\hat{R}_t = R_t$) and then repeatedly apply $\hat{R}_{t'} = r'_t + \hat{R}_{t'+1}$ backwards until $t' = t - K + 1$. We repeat this for each the input sequences to maintain the consistency of the rewards and RTGs. This process is summarised in Algorithm 2.

---

| **Algorithm 1** Relabelling return-to-go | **Algorithm 2** Generating return-to-go for DT |
|---|---|
| **Input** | **Input** |
| $\quad r_{1:T}$      rewards | $\quad r_{t-K+1:t}$ rewards |
| $\quad \hat{V}(s)$      learned value function | $\quad R_t$      return to go for time $t$ |
| $\quad T$      time horizon (trajectory length) | $\quad K$      context length |
| **Output** | **Output** |
| $\quad R_{1:T}$      relabelled return to go | $\quad \hat{R}_{1:T}$      relabelled return to go for DT |
| $R_T \leftarrow 0$ | $\hat{R}_t \leftarrow R_t$ |
| $\tau \leftarrow T$ | $\tau \leftarrow t - 1$ |
| **while** $\tau > 0$ **do** | **while** $\tau > t - K$ **do** |
| $\quad R_{\tau-1} \leftarrow r_{\tau-1} + \max(R_\tau, \hat{V}(s_\tau))$ | $\quad \hat{R}_\tau \leftarrow r_\tau + \hat{R}_{\tau+1}$ |
| $\quad \tau \leftarrow \tau - 1$ | $\quad \tau \leftarrow \tau - 1$ |
| **end while** | **end while** |

**Theoretical considerations of QDT.** Q-learning Decision Transformer (QDT) relies on DT as the agent algorithm, which can be seen as a reward conditioning model. A reward conditioning model takes the states and RTG as inputs and outputs actions. If we assume the model is trained with the state $s_t$ and the optimal state-action value function ($Q^*(s_t, a_t)$), then we can guarantee that the model will output the optimal action ($\arg\max_a Q^*(s_t, a)$) for as long as it is given $s_t$ and $\max_a Q^*(s_t, a)$ as inputs (Srivastava et al., 2019). In practice, we do not know the optimal value function $Q^*(s, a)$, hence DT (and similarly other reward conditioning approaches) uses RTG instead. RTG is collected through the behaviour policy (or policies) and often is not optimal – with the majority of values being much lower than the corresponding optimal value function ($Q^*(s, a)$). As QDT uses CQL to learn the optimal *conservative* value function, Th. 3.2 in Kumar et al. (2020) shows that the *conservative* value function is a lower bound of the true value function. Hence the QDT relabelling process moves the RTG in the training dataset closer to the optimal value function (see Appendix D).

# 4 RELATED WORK

**Offline reinforcement learning.** The offline RL learns its policy purely from a static dataset that was previously collected with an unknown behaviour policy (or policies). As the learned policy might differ from the behaviour policy, the offline algorithms must mitigate the effect of the *distributional shift* (Agarwal et al., 2020; Prudencio et al., 2022). One of the most straightforward approaches to address the issue is by constraining the learned policy to the behaviour policy (Fujimoto et al., 2019; Kumar et al., 2019a; Wu et al., 2019). Other methods constrain the learned policy by making conservative estimates of future rewards (Kumar et al., 2020; Yu et al., 2021). Some model-based methods estimate the model's uncertainty and penalize the actions whose consequences are highly uncertain (Janner et al., 2019; Kidambi et al., 2020). Some approaches address the *distributional shift* without restricting the learned policy. One such approach group is weighted imitation learning (Wang et al., 2018; Peng et al., 2019; Wang et al., 2020; Nair et al., 2020; Chen et al., 2020; Siegel et al., 2020; Brandfonbrener et al., 2021), which carries out imitation learning by putting higher weights on the good state-action pairs. It usually uses an estimated advantage function as the weight. As this approach imitates the selected parts of the behaviour policy, and it naturally restricts the learned policy within the behaviour policy. The other group of the approaches without restricting the learning policy is conditional sequence modelling, which learns a policy conditioned with a particular metric for the future trajectories. Some examples of the metrics are sum of the future rewards (Srivastava et al., 2019; Chen et al., 2021), a certain state (sub goal) (Codevilla et al., 2018; Ghosh et al., 2019; Lynch et al., 2020) and even learned features from the future trajectory (Furuta et al., 2021).

Our approach does not belong to any of these groups but is related to the approach of learning pessimistic value function, the conditional sequence modelling and weighted imitation learning approaches. Essentially, our method is a conditional sequence modelling approach as it learns the following action conditioned on the current state and the sum of the future rewards, but the training data is augmented by the result of the learned pessimistic value function. Also, the overall high-level structure is somewhat similar to the weighted imitation learning, which learns the value function and uses it to weight the training data in the following imitation learning stage. However, each component is very different from ours, and it uses the value function to weight the training data, whereas our approach relabels the RTG values by tracing back the trajectory with the learned value function as well as the trajectory itself where the learned value function is not reliable. Also, in our approach, the policy is learned with conditional sequence modelling, whereas they use non-conditional non-sequential models. We believe we can apply our relabelling approach to the weighted imitation learning algorithms, and it is an exciting avenue for future work.

**Data centric approach.** Andrew Ng recently spoke about the importance of the training data to achieve good performance from a machine learning model and suggests we should spend more of our effort on data than on the model (Data-centric Approach) (Press, 2021). He said, "In the *Data-centric Approach*, the consistency of the data is paramount and using tools to improve the data quality that will allow multiple existing models to do well." Our method can be seen as *Data-centric Approach* for offline RL, as we focus on improving the training data and using the existing models. Our method provides a tool to improve data quality.

## 5 EVALUATION

We investigate the performance of QDT relative to the offline RL algorithm with the Dynamic Programming based approach as well as the reward conditioning approach. As QDT utilises the result of CQL and it is considered as the state-of-art offline RL method, we pick CQL as the benchmark for the Dynamic Programming based approach and DT for the reward conditioning approach for the same reason. From the evaluations in this section, we would like to demonstrate the benefits and weaknesses of the Dynamic Programming approach and the reward conditioning approach and how our proposed approach (QDT) helps their weaknesses.

We start our investigation with a simple environment with sub-optimal trajectories. As it is a simple environment, a Dynamic Programming approach (CQL) should work well, and as it uses sub-optimal trajectories, the reward conditioning approach (DT) will struggle. It is interesting to see how much QDT helps in the circumstance. We also evaluate them on Maze2D environments designed to test the stitching ability with different levels of complexity. We expect that DT struggle whereas CQL and QDT performs well on them. Then, we evaluate the algorithms on complex control tasks – Open AI Gym MuJoCo environments with delayed (sparse) reward as per Chen et al. (2021). They have zero rewards at all the non-terminal states and put the total reward at the terminal state. It should make the Dynamic Programming approach (CQL) learning harder as it requires propagating the reward from the terminal state all way to the initial state. Finally, we show the evaluation results for Open AI Gym MuJoCo environments with the original dense reward setting for the reference.

**Simple environment.** To highlight the benefit of QDT, we evaluate our method in a simple environment, which has 6-by-6 discrete states and eight discrete actions. The goal of the task is to find the shortest path from the start to the goal state. We prepare an offline RL dataset with a hundred episodes from a uniformly random policy and then remove an episode that achieves close to the optimal total reward to make sure it only contains sub-optimal trajectories. Refer to Appendix B for further details of the environment and the dataset.

Table 1 show the summary of the evaluation results. We also evaluate the performance of CQL, which is used for relabeling. It shows vanilla DT fails badly, which indicates DT struggles to learn from sub-optimal trajectories, whereas CQL performs well as it employs a Dynamic Programming approach, which can pool information across trajectories and successfully figure out the near-optimal policy. It shows QDT performs similar to CQL, which indicates that although QDT uses the conditional policy approach, it overcomes its limitation and learns the near-optimal policy from the sub-optimal data. Further details and results are available in Appendix B.

Table 1: Simple Environment Evaluation Results. Average and standard deviation scores are reported over ten seeds.

|  | CQL | DT | QDT |
|---|---|---|---|
| Total Reward | **40.0** $\pm$ 0.0 | 15.9 $\pm$ 4.4 | **42.2** $\pm$ 6.3 |

**Maze2D environments.** Maze2D domain is a navigation task requiring an agent to reach a fixed goal location. The tasks are designed to provide tests of the ability of offline RL algorithms to be able to stitch together parts of different trajectories (Fu et al., 2020). It has four kinds of environments – open, umaze, medium and large, and they are getting more complex mazes in the order (Fig. 3) [2]. Also, it has two kinds of reward functions – normal and dense. The normal gives a positive reward only when the agent reaches the goal, whereas the dense gives the rewards at every step exponentially proportional to the negative distance between the agent and the goal. For the model, we use the DT source code provided by the authors [3] and d3rlpy [4] (Imai & Seno, 2021) – offline RL library for CQL, then build QDT by adding small code (replacing the return-to-go) to the DT source code before its training.

Table 2 shows the summary of the results. All of the numbers in the table are the normalised total reward (score) such that 100 represents an expert policy (Fu et al., 2020). CQL works well, espe-

---

[2]https://github.com/rail-berkeley/d4rl/wiki/Tasks
[3]https://github.com/kzl/decision-transformer
[4]https://github.com/takuseno/d3rlpy

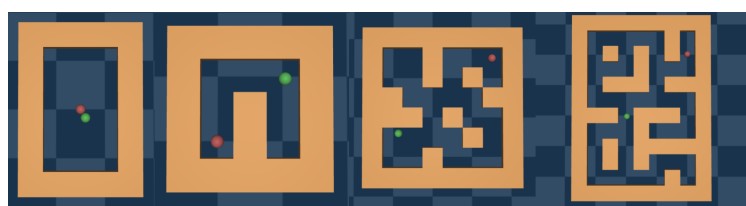

Figure 3: Four Maze2D environment layouts (from left to right: open, umaze, medium and large).

Table 2: Maze2D Evaluation Results. Average and standard deviation scores are reported over 5 seeds. The result for each seed is obtained by evaluating the last learned model on the target environment. The best average values are marked in bold.

| | Dataset | CQL | DT | QDT |
|---|---|---|---|---|
| Sparse Reward | maze2d-open-v0 | **216.7** ± 80.7 | 196.4 ± 39.6 | 190.1 ± 37.8 |
| | maze2d-umaze-v1 | **94.7** ± 23.1 | 31.0 ± 21.3 | 57.3 ± 8.2 |
| | maze2d-medium-v1 | **41.8** ± 13.6 | 8.2 ± 4.4 | 13.3 ± 5.6 |
| | maze2d-large-v1 | **49.6** ± 8.4 | 2.3 ± 0.9 | 31.0 ± 19.8 |
| Dense Reward | maze2d-open-dense-v0 | 307.6 ± 43.5 | 346.2 ± 14.3 | **325.7** ± 61.4 |
| | maze2d-umaze-dense-v1 | **72.7** ± 10.1 | −6.8 ± 10.9 | 58.6 ± 3.3 |
| | maze2d-medium-dense-v1 | **70.9** ± 9.2 | 31.5 ± 3.7 | 42.3 ± 7.1 |
| | maze2d-large-dense-v1 | **90.9** ± 19.4 | 45.3 ± 11.2 | 62.2 ± 9.9 |

cially with the dense rewards. DT struggles in many cases due to the lack of stitching ability. (These environments are designed to test the stitching ability.) QDT clearly improves DT performance, especially where CQL performs well. It indicates that QDT brings the stitching capability to DT approach. We discuss the performance gap between CQL and QDT in Section 6.

**Open AI Gym MuJoCo environments with delayed (sparse) reward.** We also evaluate our approach (QDT) on complex control tasks – Open AI gym MuJoCo environments with the D4RL offline RL datasets (Fu et al., 2020). The Open AI gym MuJoCo environments consist of three tasks *Hopper*, *HalfCheetah* and *Walker2d*. We test on *medium* and *medium-replay* v2 datasets. To demonstrate the shortcoming of the Dynamic Programing approach (CQL), we follow Chen et al. (2021) and evaluate the algorithms with a delayed (sparse) reward scenario in which the agent does not receive any reward along the trajectory and receives the sum of the rewards at the final time step. Again we use the DT and CQL models from the existing source code for the MuJoCo Gym environments without any modifications and add extra code for the relabelling of the RTG values.

Table 3 shows the simulation results (scores) for the delayed reward case. We also copy the simulation results from Chen et al. (2021) for DT and CQL for the reference. All of the numbers in the table are the normalised total reward (score) such that 100 represents an expert policy (Fu et al., 2020). As expected, CQL struggles to learn a good policy, whereas the DT shows good performance. Also, QDT performs similar to DT even though they are using the results of CQL that performs badly. It indicates that QDT successfully use the information from CQL where it is useful. One exception is the medium-replay-walker2d result. QDT performs worse than DT here. Through some investigations, we found that the CQL algorithm overestimates the value function in the majority of the states in the medium-replay-walker2d dataset. We touch the issue in the following discussion section.

**Open AI Gym MuJoCo environments.** We also evaluate our approach (QDT) on Open AI gym MuJoCo environments with the original dense reward for the reference. As they have dense rewards and contain reasonably good trajectories, both CQL and DT would work well.

Table 4 shows the summary of our simulation results for CQL, DT and QDT. We also copy the simulation results from Chen et al. (2021) for DT and Emmons et al. (2021) for CQL for the reference. Firstly, we can see that our simulation results are aligned with the references except for the medium-replay-hopper result. Because it has a relatively high variance, it is probably due to the small number of samples (five random seeds). Secondly, CQL performs equal or better than DT and QDT in this evaluation. It is understandable as they have dense rewards (they do not require propagating value function in the trajectory). Finally, from the comparison between DT and QDT, QDT performs the same as DT.

Table 3: Open AI Gym MuJoCo with Delayed Reward Evaluation Results. Average and standard deviation scores are reported with 5 seeds. Our simulation results are in the Results columns, best average boldfaced. Ref.[*2] are the results copied from Chen et al. (2021). We are not sure which version of dataset the authors used for Ref.[*2], and only *Hopper* results are available in the paper.

| | | CQL | | DT | | QDT |
|---|---|---|---|---|---|---|
| | Dataset | Results | Ref.[*2] | Results | Ref.[*2] | Results |
| **Medium** | Hopper-v2 | $23.3 \pm 1.0$ | $5.2$ | $\mathbf{57.3} \pm 2.4$ | $60.7 \pm 4.5$ | $50.7 \pm 5.0$ |
| | HalfCheetah-v2 | $1.0 \pm 1.0$ | $-$ | $42.2 \pm 0.2$ | $-$ | $\mathbf{42.4} \pm 0.5$ |
| | Walker2d-v2 | $0.0 \pm 0.4$ | $-$ | $\mathbf{69.9} \pm 2.0$ | $-$ | $63.7 \pm 6.4$ |
| **Medium Replay** | Hopper-v2 | $7.7 \pm 5.9$ | $2.0$ | $\mathbf{50.8} \pm 14.3$ | $78.5 \pm 3.7$ | $38.7 \pm 26.7$ |
| | HalfCheetah-v2 | $7.8 \pm 6.9$ | $-$ | $\mathbf{33.0} \pm 4.8$ | $-$ | $32.8 \pm 7.3$ |
| | Walker2d-v2 | $3.2 \pm 1.7$ | $-$ | $\mathbf{51.6} \pm 24.6$ | $-$ | $29.6 \pm 15.5$ |

Table 4: Open AI Gym MuJoCo Evaluation Results. Average and standard deviation scores are reported over 5 seeds. Our simulation results are in Results columns. The best average values are marked in bold. Ref.[*1] is the results copied from Emmons et al. (2021). Ref.[*2] is the results copied from Chen et al. (2021).

| | | CQL | | DT | | QDT |
|---|---|---|---|---|---|---|
| | Dataset | Results | Ref.[*1] | Results | Ref.[*2] | Results |
| **Medium** | Hopper-v2 | $\mathbf{69.4} \pm 13.1$ | $64.6$ | $60.3 \pm 5.5$ | $67.6 \pm 1.0$ | $66.5 \pm 6.3$ |
| | HalfCheetah-v2 | $\mathbf{49.2} \pm 0.5$ | $49.1$ | $42.1 \pm 0.5$ | $42.1 \pm 0.1$ | $42.3 \pm 0.4$ |
| | Walker2d-v2 | $\mathbf{83.0} \pm 0.6$ | $82.9$ | $73.3 \pm 2.5$ | $74.0 \pm 1.4$ | $67.1 \pm 3.2$ |
| **Medium Replay** | Hopper-v2 | $\mathbf{96.2} \pm 7.9$ | $97.8$ | $63.7 \pm 12.2$ | $82.7 \pm 7.0$ | $52.1 \pm 20.3$ |
| | HalfCheetah-v2 | $\mathbf{49.8} \pm 0.5$ | $47.3$ | $34.1 \pm 1.1$ | $36.6 \pm 0.8$ | $35.6 \pm 0.5$ |
| | Walker2d-v2 | $\mathbf{76.5} \pm 21.1$ | $86.1$ | $60.2 \pm 13.9$ | $66.6 \pm 3.0$ | $58.2 \pm 5.1$ |

## 6 DISCUSSION

**Stitching ability.** To demonstrate the stitching ability, we evaluate the performance of each algorithm with varying degrees of the sub-optimal dataset. We pick the medium-replay dataset for the MuJoCo Gym environment as it contains trajectories generated by various agent levels and removes the best $X\%$ of the trajectories. As $X$ is increased, more good trajectories are removed from the dataset. Thereby moving further away from the optimal setup. Fig. 4 shows the CQL, DT and QDT results as well as the best trajectory return in the dataset. It shows that CQL offers better results than the best trajectory within the dataset except $X = 0$, where the trajectory contains the best score; hence it can not be better than that. In contrast, DT fails to exceed the best trajectory, which indicates DT fails to stitch the sub-optimal trajectories. QDT performs better than DT and becomes close to the CQL results at $X = 40$ and $50$ (in the regime with $60 - 50\%$ bottom trajectories).

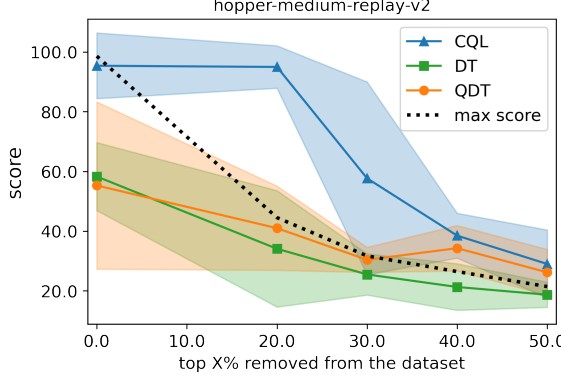

Figure 4: Evaluation results (scores) for CQL, DT and QDT with the hopper-medium-replay-v2 dataset removed top X% trajectories. The shaded area shows one standard deviation range of the results. It also has the maximum score in the dataset as a reference. CQL results are generally better than the maximum score, which indicates CQL successfully stitches sub-optimal trajectories, whereas DT fails to do so. QDT improves DT through relabelling, being better than the maximum score on the right-hand side of the plot.

**Performance gap between QDT and CQL.** Although QDT improves DT on the sub-optimal dataset scenario (Fig. 4), QDT does not perform as well as CQL for the range of small $X$. The results from Emmons et al. (2021) indicate that DT can perform as well as CQL when plenty of good trajectories are available (medium-expert dataset). It implies that there is still room for improvements for DT and QDT approaches with datasets that contain far from optimal trajectories. To address this, we are considering using a Q-learning algorithm specific to QDT approach, but this is left for the future work.

**Conservative weight.** CQL has a hyperparameter called *conservative weight*, denoted by $\alpha$ in Eq. 2. It weights the regulariser term, where the higher value, the more conservative are the value function estimations. Ideally, we would like to set it as small as possible so that the estimated value function becomes a tighter lower bound; however, too small conservative weight might break the lower bound guarantee, and the learned value function might give a higher value than the true value (Kumar et al., 2021). Empirically, we discovered that this is exactly what happens in our delayed reward experiment (Table 3) for the medium-replay-waker2d dataset example. The value function learned by CQL in the dataset has higher values than the corresponding true value in many states, and it causes the wrong relabelling of RTG and, subsequently, a worse QDT performance. We evaluated it with higher $\alpha$ values – increased from 5.0 to 100. Though this improves the QDT result from $30.3 \pm 16.2$ to $46.9 \pm 13.8$, it is still worse than DT. This is left for future work for further investigation. In this paper, we assume we have access to the environment in order to optimise the hyperparameters. However, this should be done purely offline for a proper offline RL setting. Although there are some proposals (Paine et al., 2020; Fu et al., 2021; Emmons et al., 2021), this is still an active research area.

**Reproducing results for benchmarking.** There have been many attempts to establish a benchmark for the offline RL approaches by building datasets (Fu et al., 2020; Agarwal et al., 2020), sharing their source code, as well as producing a library focusing on offline RL (Imai & Seno, 2021). However, we still found some conflicting results between papers. The leading cause of the issue is the requiring a vast amount of effort and computational power to reproduce the other researcher's results. As a result, most authors have no choice but to re-use the original results from state-of-the-art papers in the literature to establish a comparison. However, this leads to conflicting results due to the difficulties of reproducing all the details involved in these very diverse experimental setups. For example, many offline RL papers use D4RL MuJoCo datasets to evaluate their algorithms and compare them against other approaches. In this case, the datasets have three versions – namely, v0, v1 and v2. While not always clearly stated, most papers use version v0. However, some use version v2, which causes some of the conflicting results. For example, Chen et al. (2021) appears to evaluate their model with the v2 dataset while referencing other papers' results that use v0. A second issue with benchmarking the results in this manner is the usual insufficient number of simulations. As the simulations require large processing power, it is not feasible to run a large number of simulations. Most authors (including us) evaluate only three random seeds, which is often insufficient to compare the results. In this paper, we emphasise and analyse carefully the results from the simple environment, as they helps demonstrate the characteristics of the algorithm. The more complex and realistic environments are still helpful; however, the estimated variance suggest that several cases should be handled with care when extracting conclusions.

## 7 CONCLUSIONS

We proposed Q-learning Decision Transformers, bringing the benefits of Dynamic Programming (Q-learning) approaches to reward conditioning sequence modelling methods to address some of their well-known weaknesses. Our approach provides a novel framework for improving offline reinforcement learning algorithms. In this paper, to illustrate the approach, we use existing state-of-the-art algorithms for both Dynamic Programming (CQL) and reward conditioning modelling (DT). Our evaluation shows the benefits of our approach over existing offline algorithms in line with the expected behaviour. Although the results are encouraging, there is room for improvement. For example, the QDT results for Maze2D (Table 2) are better than DT but still not as good as CQL. On the other hand, the QDT results for Gym MuJoCo delayed reward (Table 3) are significantly better than CQL but not as good as DT in the walker2d environment. These need further investigation. We are also interested in trying different Dynamic Programming and reward conditioning algorithms in the proposed framework.

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

# Appendices

## A  SIMPLE ENVIRONMENT EXAMPLE TRAJECTORY DATA AND ITS COMPUTATION

This section describes the trajectory data and some computation details for the simple example shown in Fig. 1. We bring the figure here and added the state IDs in the circle (Fig. 5). The two

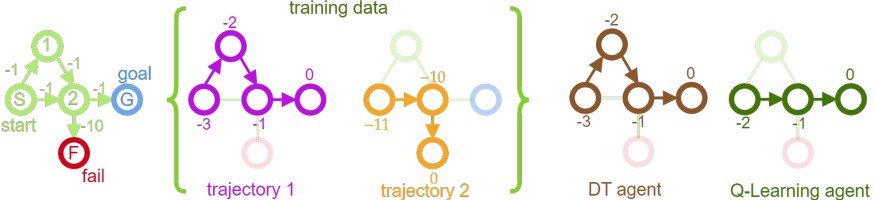

Figure 5: A simple example demonstrates the decision transformer approach's issue (lack of *stitching* ability) – fails to find the shortest path to the goal. In contrast, Q-learning finds the shortest path.

trajectories of training data are:

$$
\begin{aligned}
trajectory1 = [&s_0{=}S, a_0{=}up, r_0{=}{-}1, \\
&s_1{=}1, a_1{=}down, r_1{=}{-}1, \\
&s_2{=}2, a_2{=}right, r_2{=}{-}1, \\
&s_3{=}G, a_3{=}\text{N/A}, r_3{=}0] \\
trajectory2 = [&s_0{=}S, a_0{=}right, r_0{=}{-}11, \\
&s_1{=}2, a_1{=}down, r_1{=}{-}10, \\
&s_2{=}F, a_2{=}\text{N/A}, r_2{=}0].
\end{aligned}
\tag{3}
$$

We compute the return-to-go (RTG) from the reward $r_t$ as Eq. 4.

$$
R_t = \sum_{\tau=0}^{T} r_\tau,
\tag{4}
$$

where $R_t$ is RTG at time step $t$ and $T$ is the episode length. The trajectories with the RTGs becomes as follows:

$$
\begin{aligned}
trajectory1 = [&s_0{=}S, a_0{=}up, r_0{=}{-}1, R_0{=}{-}3, \\
&s_1{=}1, a_1{=}down, r_1{=}{-}1, R_1{=}{-}2, \\
&s_2{=}2, a_2{=}right, r_2{=}{-}1, R_2{=}{-}1, \\
&s_3{=}G, a_3{=}\text{N/A}, r_3{=}0, R_3{=}0] \\
trajectory2 = [&s_0{=}S, a_0{=}right, r_0{=}{-}1, R_0{=}{-}11, \\
&s_1{=}2, a_1{=}down, r_1{=}{-}10, R_1{=}{-}10, \\
&s_2{=}F, a_2{=}\text{N/A}, r_2{=}0, R_2{=}0].
\end{aligned}
\tag{5}
$$

DT (the reward-conditioned approach) is trained to predict actions from the state and RTG, so it takes $[s_t, R_t]$ as the input and outputs $a_t$. (Here, we assume the context length $K = 1$ for DT for simplicity.) For example, in the $t = 0$ case, the DT agent is trained to predict $a = up$ from $[s{=}S, R{=}{-}3]$ (trajectory 1) and $a = right$ from $[s{=}S, R{=}{-}11]$ (trajectory 2). For the evaluation, we set the RTG the best value ($-2$ in this case) at $t = 0$, and then the agent predicts the action from $[s{=}S, R{=}{-}2]$. Because the input $[s{=}S, R{=}{-}2]$ is closer to $[s{=}S, R{=}{-}3]$ (trajectory 1) than $[s{=}S, R{=}{-}11]$ (trajectory 2), the agent predict $a = up$ (trajectory 1) despite the optimal action is $a = right$ (trajectory 2).

# B SIMPLE ENVIRONMENT EVALUATION DETAILS

## B.1 ENVIRONMENT

The environment has 6-by-6 discrete states and eight discrete actions as shown in Fig. 6. The goal of the task is to find the shortest path from the start to the goal state. Each time step gives -10 reward and +100 reward at the goal. The optimal policy gives +50 total reward ($= 100 - 10 * 5$). We also remap the action so that the same action index is not always optimal. The mapping differs for each state but is fixed across the episodes.

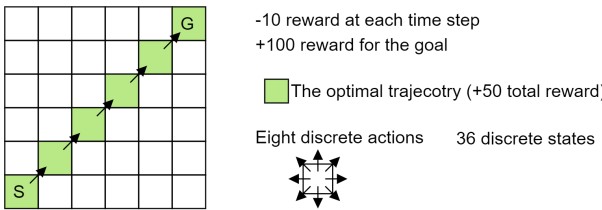

Figure 6: A simple 2D maze environment, which has 6-by-6 grid world and eight actions for moving eight directions. -10 reward at each time step and +100 reward for the goal. The optimal trajectory keeps moving up-right to the goal, which has total reward +50 ($= 100 - 10 * 5$). The action is remapped so that the same action index is not always the optimal action. The mapping differs for each state, but fixed across the episodes.

## B.2 DATASET

We prepare an offline RL dataset with a hundred episodes from a uniformly random policy and then remove an episode that achieves a positive total reward to make sure it only contains sub-optimal trajectories. As a result, the dataset used in this evaluation has one hundred episodes and 4,454 time steps. The maximum return of the hundred episodes is -10.0, the minimum return is -490 as we terminate the episode at 50 time step, and the average return is -415.5.

## B.3 CQL MODEL DETAILS

We build the CQL model for the simple environment based on Double Q-learning (Hasselt, 2010) and employ an embedding lookup table module to convert the discrete state to continuous high dimensional embedding space. The detailed model parameters are in Table 5.

Table 5: Simple Enviornment CQL Model Parameters

| Parameter | Value |
| --- | --- |
| State embedding dimension | 32 |
| DQN type | fully connected |
| DQN number of layers | 2 |
| DQN number of units | 32 |
| Optimizer | Adam |
| Optimizer betas | 0.9, 0.999 |
| Optimizer learning rate | 5.0e-4 |
| Target network update rate | 1.0e-2 |
| Batch size | 128 |
| Number of training steps | 1000 updates |
| Conservative weight ($\alpha$) | 0.5 |

### B.4 DT AND QDT MODEL DETAILS

Our DT and QDT model for the simple environment is constructed based on minGPT open-source code[5]. The detailed model parameters are in Table 6.

Table 6: Simple Environment DT/QDT Model Parameters

| Parameter | Value |
|---|---|
| Number of layers | 4 |
| Number of attention heads | 4 |
| Embedding dimension | 64 |
| Nonlinearity function | ReLU |
| Batch size | 64 |
| Context length $K$ | 2 |
| return-to-go conditioning | 50 |
| Dropout | 0.1 |
| Learning rate | 4.0e-4 |

### B.5 FURTHER EVALUATION RESULTS FOR SIMPLE ENVIRONMENT

The following tables have the simple environment results for all ten seeds. Table 7 shows the reward for the highest value during the training period. Table 8 shows the reward with the model at the end of training. DT and QDT have more significant differences between these two tables than the CQL results, which indicates that DT and QDT have overfitting issues and unstable learning behaviour.

Table 7: Simple Environment Full Results (Best). The results from the best performing model during the training.

| | CQL | DT | QDT |
|---|---|---|---|
| | 40.0 | 18.2 | 43.6 |
| | 40.0 | 20.4 | 42.0 |
| | 40.0 | 11.2 | 49.2 |
| | 40.0 | 13.8 | 42.6 |
| results for ten random seeds | 40.0 | 12.6 | 39.2 |
| | 40.0 | 8.4 | 27.8 |
| | 40.0 | 19.6 | 47.2 |
| | 40.0 | 21.2 | 47.4 |
| | 40.0 | 14.4 | 37.4 |
| | 40.0 | 18.8 | 46.0 |
| mean | 40.0 | 15.9 | 42.2 |
| std. | 0.0 | 4.4 | 6.3 |

## C OPEN AI GYM MUJOCO AND MAZE2D EVALUATION DETAILS

### C.1 CQL MODEL DETAILS

For MuJoCo Gym CQL evaluation, we use d3rlpy library (Imai & Seno, 2021). It provides a script to run the evaluation (d3rlpy/reproduce/offline/cql.py), and it uses the same hyperparameters as

---

[5]https://github.com/karpathy/minGPT

Table 8: Simple Environment Full Results (Last). The results from the model at the end of the training.

|  | CQL | DT | QDT |
|---|---|---|---|
| | 40.0 | -39.2 | 13.8 |
| | 40.0 | 8.6 | 35.8 |
| | 40.0 | -25.4 | 46.6 |
| | 40.0 | -20.8 | 16.6 |
| results for ten random seeds | 30.0 | -50.2 | 29.2 |
| | 40.0 | -26.0 | 19.6 |
| | 40.0 | 9.4 | 44.0 |
| | 30.0 | -35.0 | 47.4 |
| | 40.0 | -10.2 | 23.2 |
| | 40.0 | 7.8 | 35.0 |
| mean | 38.0 | -18.1 | 31.1 |
| std. | 4.2 | 21.3 | 12.5 |

Kumar et al. (2020). For Mazed2d simulations, we re-use the same d3rlpy script with the same hyperparameter settings.

## C.2 DT AND QDT MODEL DETAILS

For DT simulations, we use the code provided by the original paper authros[6] for both MuJoCo Gym and Maze2D environments. For QDT simulations, we added extra code to relabelling the return-to-go to the DT script (decision-transformer/gym/experiment.py). The relabelling code is described in Algorithm 1 and 2.

## C.3 EVALUATION PROCESS

**CQL** We train the CQL model with five random seeds for 500,000 updates with 256 batch size, then evaluate the model at the end of the training with 10 episode roll-outs. We inherit these CQL settings from d3rlpy offline RL library (Imai & Seno, 2021).

**DT** We train the DT model with five random seeds for 100,000 updates with 64 batch size, then evaluate the model at the end of the training with 100 episode roll-outs. We inherit these DT settings from the source code provided by the DT paper authors[7] (Chen et al., 2021).

**QDT** We train the QDT model with five random seeds, each of them employing its own trained CQL model to relabel the dataset. QDT model is trained for 100,000 updates for MuJoCo Gym and 150,000 updates for maze2d with 64 batch size, then evaluate the model at the end of the training with 100 episode roll-outs – same as DT.

## C.4 HYPER PARAMETER SEARCH

We use the same hyper-parameter settings as the original papers (Kumar et al., 2020; Chen et al., 2021). However, we did some hyper-parameter searches for the conservative weight ($\alpha$). It is because the optimal conservative weight value could be different for CQL and QDT.

For MuJoCo Gym environments, we start with $\alpha = 10.0$ for medium dataset and $\alpha = 5.0$ for medium-replay dataset. We take these values from the CQL paper. Then, reduce these values to see if the performance of CQL and QDT varies. Table 9 and Table 10 shows CQL and QDT results

---

[6]https://github.com/kzl/decision-transformer
[7]https://github.com/kzl/decision-transformer

respectively. These results show that $\alpha = 10.0$ for medium dataset and $\alpha = 5.0$ for medium-replay dataset perform well for QDT and do not degrade performance significantly for CQL. Also, they are the same values as the original paper, so we decide to keep them the same as the paper.

Table 9: CQL results for Open AI Gym MuJoCo with conservative weight parameter ($\alpha$) sweep. Average and standard deviation scores are reported over three seeds.

| | Dataset | CQL | | | |
| | | $\alpha = 10.0$ | $\alpha = 5.0$ | $\alpha = 2.5$ | $\alpha = 1.25$ |
|---|---|---|---|---|---|
| Medium | Hopper-v2 | $68.7 \pm 16.4$ | $72.5 \pm 9.5$ | $\mathbf{83.6} \pm 3.8$ | |
| | HalfCheetah-v2 | $48.9 \pm 2.4$ | $51.8 \pm 2.4$ | $\mathbf{57.0} \pm 1.1$ | |
| | Walker2d-v2 | $83.3 \pm 0.5$ | $\mathbf{86.2} \pm 0.5$ | $43.5 \pm 43.6$ | |
| Medium Replay | Hopper-v2 | | $\mathbf{95.4} \pm 11.6$ | $87.5 \pm 24.7$ | $90.7 \pm 14.5$ |
| | HalfCheetah-v2 | | $49.9 \pm 2.9$ | $51.8 \pm 2.7$ | $\mathbf{54.3} \pm 0.2$ |
| | Walker2d-v2 | | $\mathbf{88.9} \pm 3.7$ | $50.6 \pm 36.3$ | $16.8 \pm 14.2$ |

Table 10: QDT results for Open AI Gym MuJoCo with conservative weight parameter ($\alpha$) sweep. Average and standard deviation scores are reported over three seeds.

| | Dataset | QDT | | | |
| | | $\alpha = 10.0$ | $\alpha = 5.0$ | $\alpha = 2.5$ | $\alpha = 1.25$ |
|---|---|---|---|---|---|
| Medium | Hopper-v2 | $\mathbf{68.6} \pm 7.5$ | $65.3 \pm 1.3$ | $57.5 \pm 6.6$ | |
| | HalfCheetah-v2 | $\mathbf{42.2} \pm 0.5$ | $\mathbf{42.2} \pm 0.05$ | $42.1 \pm 0.4$ | |
| | Walker2d-v2 | $65.9 \pm 3.6$ | $\mathbf{70.1} \pm 2.4$ | $68.8 \pm 6.9$ | |
| Medium Replay | Hopper-v2 | | $55.3 \pm 28.0$ | $40.2 \pm 5.9$ | $\mathbf{64.0} \pm 22.9$ |
| | HalfCheetah-v2 | | $\mathbf{35.7} \pm 0.6$ | $35.5 \pm 0.4$ | $33.0 \pm 0.5$ |
| | Walker2d-v2 | | $59.1 \pm 2.8$ | $\mathbf{64.3} \pm 5.9$ | $45.2 \pm 39.5$ |

For maze2d environment, we start with $\alpha = 10.0$ which is the value used in the CQL paper for MuJoCo Gym environments medium datasets. Then, reducing these values to see if the performance of CQL varies. Table 11 shows the simulation results. We pick $\alpha = 1.0$ as it performs the best. It is possible that even lower values might perform better. We see QDT shows good improvement over DT with $\alpha = 1.0$, so we use the value for this paper. We would like to try further optimisation in the future.

Table 11: CQL results for Maze2D with conservative weight parameter ($\alpha$) sweep. Average and standard deviation scores are reported over three seeds.

| | CQL | | |
| Dataset | $\alpha = 10.0$ | $\alpha = 2.0$ | $\alpha = 1.0$ |
|---|---|---|---|
| maze2d-umaze-v1 | $27.3 \pm 12.2$ | $66.1 \pm 9.8$ | $\mathbf{96.0} \pm 32.2$ |
| maze2d-medium-v1 | $-3.5 \pm 1.3$ | $\mathbf{36.6} \pm 3.7$ | $35.9 \pm 15.3$ |
| maze2d-large-v1 | $-2.5 \pm 0.0$ | $40.8 \pm 6.0$ | $\mathbf{53.2} \pm 7.0$ |

# D  JUSTIFICATION OF REPLACING RTG WITH THE LEARNED VALUE FUNCTION

Define the optimal state value function as $V^*(s_t)$, the learned lower bound of the value function as $\hat{V}(s_t)$ and the corresponding return-to-go value as $R_t$. We show that when $\hat{V}(s_t) > R_t$, the error in $\hat{V}(s_t)$ is smaller than the error in $R_t$. We start from the condition,

$$\hat{V}(s_t) > R_t$$
$$V^*(s_t) - \hat{V}(s_t) < V^*(s_t) - R_t. \tag{6}$$

As $\hat{V}(s_t)$ is the lower bound of $V^*(s_t)$, $V^*(s_t) \geq \hat{V}(s_t)$. Hence both sides of the above equation are non-negative. We can take the absolute of both terms, and we get,

$$|V^*(s_t) - \hat{V}(s_t)| < |V^*(s_t) - R_t|. \tag{7}$$

This indicates that the error in $\hat{V}(s_t)$ is smaller than the error in $R_t$.

# E   FURTHER DISCUSSIONS

## E.1   WHY CQL OUTPERFORMS DT/QDT ON MAZE2D, BUT FAILS ON MUJOCO GYM DELAYED REWARD?

It is because maze2d are simpler environments and have shorter episodes than the MuJoCo control tasks. Table 12 shows that the action dimension, the state (observation) dimension and the episode length averaged over the top 5% returns in the dataset. It can be seen that MuJoCo tasks have higher action/state dimensions and longer episode lengths than Maze2d. Also, the evaluation results for the Sparse maze2d-medium and -large show some notable performance loss against the Dense counterparts, which is aligned with the fact that their episode lengths are longer than the maze2d-open and -umaze.

Table 12: MuJoCo Gym and Maze2D environments comparison. The table shows that the action dimension, the state (observation) dimension and the episode length averaged over the top 5% returns in the dataset.

| Environment | Action Dimension | State Dimension | Good Episode Average Length |
|---|---|---|---|
| hopper | 3 | 11 | 708.2 |
| halfcheetah | 6 | 17 | 1000.0 |
| walker2d | 6 | 17 | 996.7 |
| maze2d-open | 2 | 4 | 49.8 |
| maze2d-umanze | 2 | 4 | 128.6 |
| maze2d-medium | 2 | 4 | 224.1 |
| maze2d-large | 2 | 4 | 314.6 |

## E.2   WHY QDT OUTPERFORMS DT ON MAZE2D WHEREAS IT DOES NOT ON GYM DESPITE BOTH HAVING DENSE REWARDS?

It is due to the difference in the training data. maze2d dataset is designed to test the stitching ability; hence it only has sub-optimal trajectories, whereas the MuJoCo Gym dataset has some optimal trajectories. If the dataset has some optimal trajectories, DT will perform well. On the other hand, if the dataset has only suboptimal trajectories, DT will struggle, and QDT improves such cases by utilising the information in CQL.

As maze2d only has suboptimal trajectories, DT struggles with them, and QDT can perform better than DT. For MuJoCo Gym cases, the dataset has some optimal trajectories; hence DT performs well, and so as QDT.

Strictly speaking, there are some exceptions. MuJoCo halfcheetah-medium and halfcheetah-medium-replay dataset does not have an optimal trajectory, still QDT performs similarly to DT. It is because even CQL struggles to achieve good performance on these datasets. (CQL only performs similarly to DT even though CQL can stitch the suboptimal trajectories.) As CQL struggled, QDT could not get much help from CQL.

The other exception is maze2d-open and maze2d-open-dense. These datasets have good trajectories. It is actually aligned with our evaluation results. The results for maze2d-open and maze2d-open-dense show good performance with DT.

Table 13 shows the maximum, 95 percentile and 90 percentile values of the normalised returns (score) in the dataset. As we discussed above, Maze2d has suboptimal trajectories (except open

and open-dense), and MuJoCo Gym has (near) optimal trajectories – a score close to 100 (except halfcheetah).

Table 13: Scores in MuJoCo Gym and Maze2D datasets. This table shows that maximum score, 95 percentile score and 90 percentile score values for each dataset.

| Dataset | max. score | 95 pct. score | 90 pct. score |
|---|---|---|---|
| maze2d-open-v0 | 232.4 | 130.7 | 116.2 |
| maze2d-open-dense-v0 | 188.9 | 128.4 | 117.4 |
| maze2d-umaze-v1 | 21.1 | 13.2 | 10.3 |
| maze2d-umaze-dense-v1 | -1.4 | -11.7 | -18.3 |
| maze2d-medium-v1 | 12.8 | 6.8 | 4.9 |
| maze2d-medium-dense-v1 | 8.9 | 4.0 | 0.3 |
| maze2d-large-v1 | 16.9 | 6.5 | -2.5 |
| maze2d-large-dense-v1 | 14.6 | 7.9 | -2.4 |
| hopper-medium-v2 | 99.5 | 63.2 | 57.0 |
| hopper-medium-replay-v2 | 98.6 | 46.4 | 31.5 |
| halfcheetah-medium-v2 | 45.0 | 43.0 | 42.5 |
| halfcheetah-medium-replay-v2 | 42.4 | 39.9 | 39.2 |
| walker2d-medium-v2 | 92.0 | 83.4 | 82.4 |
| walker2d-medium-replay-v2 | 89.9 | 66.6 | 42.5 |

### E.3 Why QDT performs close to DT, not CQL in Fig. 4 (Gym hopper)?

The main reason is that QDT employs DT as its agent algorithm. The difference lays in its training data. If the environment/dataset has specific characteristics that work against DT approach, those also work against QDT. Some of these properties, such as dataset sub-optimality, are fixed/mitigated by QDT. However, there may be other elements that are against DT and QDT, e.g., the environment having a few critical states (Kumar et al., 2022). If this is also behind the gap between CQL and DT, then it is possible QDT performs close/the same as DT.

Kumar et al. (2022) studied the Dynamic Programming approach and the imitation learning approach and compared the upper bounds of their sub-optimality (the difference between the return from the optimal policy and the learned policy). They show that the Dynamic Programming approach is preferred over imitation learning when the environment has a few critical states – the return of the episode mostly depends upon the actions in these states. The results in Kumar et al. (2022) are based on theoretical analysis (sub-optimality upper bounds). Hence, it is possible that the imitation learning approach (DT and QDT) can perform as well as or better than Dynamic Programming approaches (such as CQL) in practice. Kumar et al. (2022) empirically shows that the goal-conditioned approach remains competitive by selecting the right level of model capacity and the goal. There are still many open and ongoing discussions regarding the comparison.

### E.4 Extra results for removing top X%

We run the same experiment as the Stitching ability subsection in Section 6 on the other two MuJoCo Gym environments. The results (Fig. 7) do not show a clear benefit of QDT over DT. We think it is because the cause of the gap between CQL and DT is not just the sub-optimality in the dataset (still the sub-optimality can be the cause of the difference, but it is not the only cause in these cases.)

### E.5 Consistency relabelling ablation experiment

We have tried the ablation experiment for the consistency relabelling (Algorithm 2) on a subset of environments. The results are summarised it in Table 14. We run ten random seeds for Simple Environment and three for others. The Simple Environment result shows some benefits of using Algorithm 2 in its average value, although it is not significant. For the other more complex environments, we do not see clear benefits of Algorithm2. We think this is because the changes applied by Algorithm2 are relatively minor compared to the original RTG variations. We think it is better to

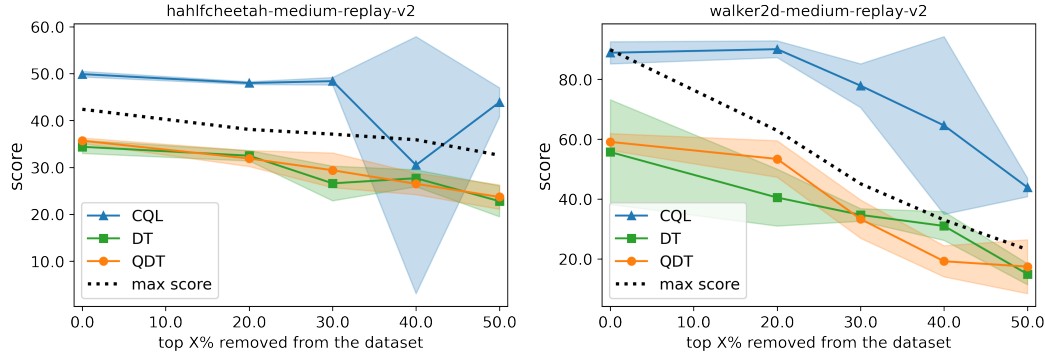

Figure 7: Evaluation results (scores) for CQL, DT and QDT with the halfcheetah-medium-replay-v2 and walker2d-medium-replay-v2 dataset removed top X% trajectories. The shaded area shows one standard deviation range of the results. It also has the maximum score in the dataset as a reference.

keep Algorithm 2, at least for now because the training data could have non-realistic (inconsistent) samples without the algorithm.

Table 14: Scores in Simple Environment, MuJoCo Gym and Maze2D datasets. This table shows QDT results and QDT without the consistency relabelling (Algorithm 2).

| Dataset | QDT | QDT w/o Alg.2 |
|---|---|---|
| Simple Environment | $42.2 \pm 6.3$ | $29.7 \pm 13.8$ |
| hopper-medium-v2 | $65.3 \pm 2.0$ | $65.7 \pm 3.9$ |
| halfcheetah-medium-v2 | $42.2 \pm 2.3$ | $42.4 \pm 0.1$ |
| walker2d-medium-v2 | $71.3 \pm 2.4$ | $80.2 \pm 10.8$ |
| maze2d-large-v1 | $35.0 \pm 24.2$ | $23.0 \pm 5.0$ |

### E.6 AGGREGATED EVALUATION RESULTS

We compute the aggregated evaluation results for each group of environments (maze2d, MuJoCo Gym delayed reward and MuJoCo Gym) with three different metrics – median, Interquantile mean (IQM) and mean (Fig. 8). It uses 95% stratified bootstrapped confidence interval (Agarwal et al., 2021).

The results support our conclusions 1) DT struggles in maze2d, but QDT improves DT performance by getting help from CQL. 2) CQL fails in MuJoCo Gym delayed reward. 3) DT and QDT perform similarly in MuJoCo Gym. Note that QDT has never failed in any of these groups of environments.

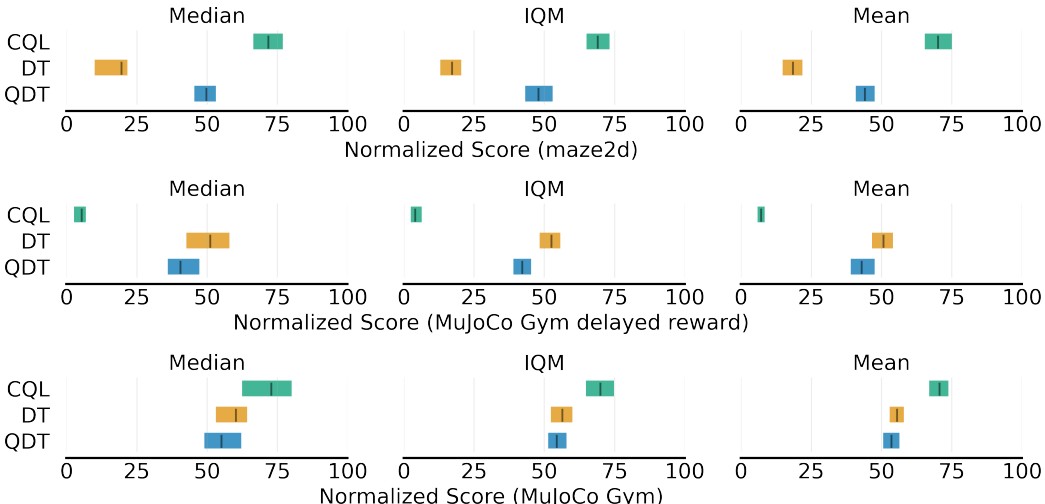

Figure 8: Aggregated evaluation results (scores) for each group of environments (maze2d, MuJoCo Gym delayed reward and MuJoCo Gym) with three different metrics – median, Interquantile mean (IQM) and mean. The results support our conclusions 1) DT struggles in maze2d, but QDT improves DT by getting help from CQL. 2) CQL fails in MuJoCo Gym delayed reward. 3) DT and QDT perform similarly in MuJoCo Gym.

