# OpenReview forum: "Q-learning Decision Transformer: Leveraging Dynamic Programming for Conditional Sequence Modelling in Offline RL"
_ICLR.cc/2023/Conference — Submitted to ICLR 2023_

### Official Review · Reviewer_C8yF · 2022-10-25

**Confidence:** 3
**Correctness:** 2
**Technical Novelty And Significance:** 2
**Empirical Novelty And Significance:** 2
**Recommendation:** 3

**Clarity, Quality, Novelty And Reproducibility:**

As far as I know, this work is original. The paper is written clearly, but the presentation can be further improved.

**Strength And Weaknesses:**

This manuscript studies an interesting question for DT and offline RL. The idea of RTG relabelling is simple, the resulting algorithm is easy to implement. It does improve DT's performance on several antmaze tasks.

Overall, I see this is an interesting idea, yet I think further explorations can be made in a few directions to make the current manuscript a strong submission. See discussions below.

*  What the authors want to show in Figure 1 is that trajectory 1 and 2 have very different and even polarized RTGs, when conditioning on a reasonable RTG (which can be out of distribution), DT might predict the suboptimal move at the initial state.

A minor point I want to mention --  the example trajectory 2 in Fig 1, actually suggests the MDP the authors consider is not episodic. For continuing tasks/MDPS, people tend to use discounted rewards, and the RTG would be finite (assuming bounded reward).  Assuming episodic MDP (a finite horizon or a terminating state), trajectory 2 wouldn't contain any infinite RTG either. The original paper of DT actually implicitly assumed episodic MDP. I believe the above idea can be illustrated with examples with finite RTGs as well.

* Given this example, the central question to answer becomes how to choose the right move if the optimal subtrajectory is associated with suboptimal RTG. I can see the problem can be generalized to if the return distribution of the offline dataset contains different modes, how to stitch suboptimal trajectories associated with trajectories in different modes. The authors propose a DP approach to relabel the RTGs. I suggest the authors present the algorithms in the main paper for clarity, also use the concise math notations like :  R_{t-1} = max(R_t, Vhat(s_t)) + r_{t_1}, where Vhat is the estimated value function.

One particular design I noticed is that for each input sequence, the RTG is further relabelled as in Algorithm 2. The authors argue this is to keep the consistency between the reward and RTG within an input sequence. I'd suggest the authors run an ablation experiment to verify this is needed, and give some explanations. Will the performance of DT crash with such inconsistency?

Besides, this means, for the same state action tuple (s_t, a_t), the associated RTG will be different when it is sampled in different input sequences. I wonder if this is related to data augmentation / perturbation -- you smooth out the input data distribution by those "noisy" RTGs, and I wonder how much performance gain will you obtain by introducing this extra labelling step.

* It's better to give more details for the experiments, e.g. - how many evaluation rollouts per seed are taken? Besides, each algorithm only takes 3 seeds. Preferably, the authors can run experiments with 5 to 10 seeds. (The authors can also use the stratified bootstrap to compute statistics rather than mean: https://arxiv.org/pdf/2108.13264.pdf )

* For the experiments, the proposed QDT improves upon DT on some antmaze tasks, but the performance is roughly the same as DT on the mujoco tasks (both the delayed reward mode and the normal mode). It suggests the improvement is marginal. For the ablation experimment shown in Fig 4,  DT and QDT perform all most the same.  I'm also curious why CQL works for antmaze with sparse rewards, but not Gym tasks with delayed reward?

The results are also somewhat inconsistent. QDT improves upon DT for antmaze tasks with dense rewards, but the performance are roughly the same for Gym tasks whose rewards are also dense. I'd suggests the authors try to answer why such inconsistency happens.

**Summary Of The Paper:**

The authors have observed that DT (for offline RL) lacks the ability to stitch suboptimal sub-trajectories, and propose the following two-step procedure:

1) learn a Q function using the CQL critic
2) relabel the RTG of the offline dataset using the learned critic, then train a DT on the relabelled dataset


**Summary Of The Review:**

A paper with interesting idea but lacks ablation study. The experiment results show some inconsistent patterns and it is unclear the proposed method can perform consistently better than state-of-the-art.

---

> ### Author Response · Authors · 2022-11-15
> **Reply to Reviewer C8yF (2)**
>
> > I'm also curious why CQL works for antmaze with sparse rewards, but not Gym tasks with delayed reward?
>
> (In this response, we assume the reviewer means maze2d instead of antmaze.)
>
> It is because maze2d includes simpler environments that have shorter episodes than the MuJoCo control tasks. Please refer to the following table:
>
> | Environment | Action Dim. | State Dim. | Ave. Good Episode Len. |
> | ----------- | :---------: | :--------: |----------------------- |
> | hopper      | 3           | 11         | 708.2 |
> | halfcheetah | 6           | 17         | 1000.0 |
> | walker2d    | 6           | 17         | 996.7 |
> ||||
> | maze2d-open | 2           | 4          | 49.8 |
> | maze2d-umanze | 2         | 4          | 128.6 |
> | maze2d-medium | 2         | 4          | 244.1 |
> | maze2d-large  | 2         | 4          | 314.6 |
> ||||
>
> This table shows that the action dimension, the state (observation) dimension and the episode length averaged over the top 5% returns in the dataset. As seen in the table, MuJoCo tasks have higher action/state dimensions and longer episode lengths than Maze2d. Also, the evaluation results for the Sparse maze2d-medium and -large show some notable performance loss against the Dense counterparts, which is aligned with the fact that their episode lengths are longer than the maze2d-open and -umaze.
>
> We have added this discussion to Appendix E.1.
>
> > The results are also somewhat inconsistent. QDT improves upon DT for antmaze tasks with dense rewards, but the performance are roughly the same for Gym tasks whose rewards are also dense. I'd suggests the authors try to answer why such inconsistency happens.
>
> (In this response, we assume the reviewer means Maze2d instead of antmaze.)
>
> This is due to differences in the training data, namely, the maze2d dataset is designed to test the stitching ability; hence it only has suboptimal trajectories, whereas the MuJoCo Gym dataset has some optimal trajectories. If the dataset includes some optimal trajectories, DT will perform well. On the other hand, if the dataset contains only suboptimal trajectories, DT will struggle, while QDT will work well in such cases by utilising the information from CQL.
>
> As maze2d only includes suboptimal trajectories, DT struggles with them -- and QDT can perform better than DT. For the MuJoCo Gym cases, the dataset has some optimal trajectories; hence DT performs well, and so does QDT.
>
> Strictly speaking, there are some exceptions. Although the MuJoCo halfcheetah-medium and halfcheetah-medium-replay datasets do not have an optimal trajectory, still QDT performs similarly to DT. It is because even CQL struggles to achieve good performance on these datasets (CQL only performs similarly to DT even though CQL can stitch the suboptimal trajectories). When CQL struggles, QDT could not leverage its access to CQL's information.
>
> The other exception comes from maze2d-open and maze2d-open-dense. Both these datasets offer good trajectories. This is actually aligned with our evaluation results. The results for maze2d-open and maze2d-open-dense show good performance with DT.
>
> The following table shows the maximum, 95 percentile and 90 percentile values of the normalised returns (score) in the datasets. As we discussed above, Maze2d has suboptimal trajectories (except open and open-dense), and MuJoCo Gym has (near) optimal trajectories -- a score close to 100 (except halfcheetah).
>
> | dataset | max. return | 95 percentile | 90 percentile |
> | ------- | :---------: | :-----------: | :-----------: |
> |maze2d-open-v0      | 232.4 | 130.7 | 116.2 |
> |maze2d-open-dense-v0| 188.9 | 128.4 | 117.4 |
> |maze2d-umaze-v1     | 21.1  | 13.2  | 10.3  |
> |maze2d-umaze-dense-v1| -1.4 | -11.7 | -18.3 |
> |maze2d-medium-v1    | 12.8  | 6.8   | 4.9   |
> |maze2d-medium-dense-v1|8.9  | 4.0   | 0.3   |
> |maze2d-large-v1     | 16.9  | 6.5   | -2.5  |
> |maze2d-large-dense-v1| 14.6 | 7.9   | -2.4  |
> ||||
> |hopper-medium-v2        | 99.5 | 63.2 | 57.0  |
> |hopper-medium-replay-v2 | 98.6 | 46.4 | 31.5  |
> |halfcheetah-medium-v2   | 45.0 | 43.0 | 42.5  |
> |halfcheetah-medium-replay-v2 |42.4 | 39.9 | 39.2 |
> |walker2d-medium-v2      | 92.0 | 83.4 | 82.4 |
> |walker2d-medium-replay-v2 | 89.9 | 66.6 | 42.5 |
> ||||
>
> We have added this discussion to Appendix E.2.
>
> We thank the reviewer for providing such detailed feedback and interesting thoughts and we hope we have managed to addressed the different points they raised. We believe at the community at ICLR would benefit from seeing our proposed approach and inspire research into more complex combinations between existing RL algorithms.

---

> ### Author Response · Authors · 2022-11-15
> **Reply to Reviewer C8yF (1)**
>
> > A minor point I want to mention -- the example trajectory 2 in Fig 1, actually suggests the MDP the authors consider is not episodic. For continuing tasks/MDPS, people tend to use discounted rewards, and the RTG would be finite (assuming bounded reward).
>
> Thank you for raising this point, as it is an important detail.
> We have incorporated the suggestion and have changed the model in Fig.1 to an episodic task by introducing a "fail" state - i.e. the terminal state. This prevents the trajectory to continue infinitely. This change has been introduced in the latest version of the paper.
>
> > I suggest the authors present the algorithms in the main paper for clarity, also use the concise math notations like : R_{t-1} = max(R_t, Vhat(s_t)) + r_{t_1}, where Vhat is the estimated value function.
>
> Thank you for the suggestion. We have updated the notation accordingly in the algorithm and moved it into the main body of the paper.
>
> > One particular design I noticed is that for each input sequence, the RTG is further relabelled as in Algorithm 2. The authors argue this is to keep the consistency between the reward and RTG within an input sequence. I'd suggest the authors run an ablation experiment to verify this is needed, and give some explanations. Will the performance of DT crash with such inconsistency?
>
> This is indeed a very interesting experiment. We have tried it on a subset of environments and summarised the main results in the table below. We ran ten random seeds for Simple Environment and three for others. The Simple Environment result indicates potential benefits of using Algorithm 2 in its mean value, but for the other more complex environments, we do not see a clear trend that supports Algorithm2's added value. We argue this is because the changes introduced in Algorithm2 are relatively minor compared to the original RTG variations in most cases. However, its main benefit is to prevent non-realistic examples.
>
> | Dataset | QDT | QDT w/o Algorithm2 |
> | ------- | :-: | :----------------: |
> |Simple Environment   | $42.2\pm6.3$ | $29.7\pm13.8$|
> ||||
> |hopper-medium-v2     | $65.3\pm2.0$ | $65.7\pm3.9$ |
> |halfcheetah-medium-v2| $42.2\pm2.3$ | $42.4\pm0.1$ |
> |walker2d-medium-v2   | $71.3\pm2.4$ | $80.2\pm10.8$|
> ||||
> |maze2d-large-v1      | $35.0\pm24.2$| $23.0\pm5.0$ |
> ||||
>
>
> > I wonder if this is related to data augmentation / perturbation -- you smooth out the input data distribution by those "noisy" RTGs, and I wonder how much performance gain will you obtain by introducing this extra labelling step
>
> This is an interesting thought. In our view, QDT does more than smoothing the noisy RTGs. In the case of a suboptimal dataset, QDT changes the RTGs to the RTG that the optimal policy could achieve (where CQL learns the value function correctly). As a result, QDT adds the stitching ability to the DT agent. In general, it might be appealing to consider data augmentation/perturbation for offline RL.
>
> With regards to the second point, could we ask the reviewer to clarify the meaning of "this extra labelling step"? We are wondering if they refer to the QDT relabelling or a different relabelling method.
>
> > It's better to give more details for the experiments, e.g. - how many evaluation rollouts per seed are taken?
>
> Thank you for the suggestion. We added all experimental details (batch size, update steps and number of roll-outs) in Appendix C.3.
>
> > Besides, each algorithm only takes 3 seeds. Preferably, the authors can run experiments with 5 to 10 seeds. (The authors can also use the stratified bootstrap to compute statistics rather than mean: https://arxiv.org/pdf/2108.13264.pdf )
>
> We ran further experiments using different random seeds (five seeds in total) and updated Table2, 3 and 4 in the paper. Also, we computed the stratified bootstrap based on the suggested paper and put the results in Appendix E.6. The results are in line with the results in Table2, 3 and 4.

---

### Official Review · Reviewer_QqTj · 2022-10-26

**Confidence:** 4
**Correctness:** 4
**Technical Novelty And Significance:** 2
**Empirical Novelty And Significance:** 3
**Recommendation:** 6

**Clarity, Quality, Novelty And Reproducibility:**

The paper is clear and of relatively high quality. The author is confident the result can be reproduced.

**Strength And Weaknesses:**

Strength:
- This is a very timely paper. Augmenting Decision Transformer is a very trendy area at this moment -- using $V(s)$ to relabel RTG is a very straightforward idea, but this is the first paper to do it. Not only did they implement the change, but they also showed a comprehensive set of experiments AND motivated the change well. I find it impressive to pull this off in such a short time.
- This paper is not afraid of revealing QDT's weakness. There are quite a few experiments and discussions that show that QDT does not quite work (yet). This might be a problem if this isn't such a fast-moving area of research -- since this area is very hot, any kind of new insight is valuable. The authors clearly know this and aren't afraid of sharing these negative results on their own algorithm. This will allow fast iteration of ideas and future research to build on these observations.
- CQL overestimates Q values (despite conservatism). This problem has been pointed out in [1], and I'm also glad the author dedicated a whole paragraph to discuss this and offered a small ablation experiment.

Weakness:
- I wish more thought could be put into the algorithm design and address the weaknesses/issues discussed. But I also understand there isn't enough time to get these done.
- QDT essentially only works (outperforms DT) in 1 environment (aka Maze2D). So, even though the idea is cool and well-motivated, there's not enough evidence that this is the **right** algorithm. But then again, research is iterative, and future papers will probably get it right.

[1] A Workflow for Offline Model-Free Robotic Reinforcement Learning

**Summary Of The Paper:**

This paper tries to unify Q-learning and Decision Transformer (imitation learning), which hopes to allow DT to obtain some level of trajectory stitching capabilities.

**Summary Of The Review:**

If the Decision Transformer were not a hot topic at this very moment (2022 Fall), this paper should be rejected because although the idea is novel, it's a bit simplistic and Important weaknesses haven't been fully addressed. However, because this field is very hot and fast-moving, we should reward fast workers -- people who can try out ideas and offer interesting insights (even if their idea "fails"). In this case, I would consider QDT a failed attempt but a very interesting failed attempt. Although this is the direction we all want to go, QDT isn't the right approach. In the spirit of 1). Rewarding negative results (and the honesty of the authors); 2). Encouraging more work in this fast-moving area; 3). Reward how well-written this paper is and how insightful the discussions/experiments are; I recommend this paper be accepted.

---

> ### Author Response · Authors · 2022-11-15
> **Reply to Reviewer QqTj**
>
> We thank the reviewer for the positive feedback and encouraging comments. Please find below our responses to the points raised.
>
> > CQL overestimates Q values (despite conservatism). This problem has been pointed out in [1] , and I'm also glad the author dedicated a whole paragraph to discuss this and offered a small ablation experiment
>
> Thank you for pointing us towards [1]. It is indeed very relevant, and as such, we added the citation to our paper.
>
> > I wish more thought could be put into the algorithm design and address the weaknesses/issues discussed. But I also understand there isn't enough time to get these done.
> > QDT essentially only works (outperforms DT) in 1 environment (aka Maze2D). So, even though the idea is cool and well-motivated, there's not enough evidence that this is the right algorithm. But then again, research is iterative, and future papers will probably get it right.
>
> We intended to evaluate the algorithms in various environments and show each algorithm's strong and weak points. Although QDT does not outperform the other approaches in all cases, it avoids the worst failures (DT fails in maze2d and CQL fails in MuJoCo delayed reward).
>
> We agree that there is room for several improvements. It is our intention that this paper provides a baseline for this novel approach, as the first work combining CQL and DT. Whilst we agree that there are alternatives for this combination, we believe that we can use this work to draw the ICLR community's attention to the issues presented in the paper, as they are pervasive (and often overlooked) in the literature.
> Again, we appreciate the reviewer's understanding of the paper's value.

---

### Official Review · Reviewer_J4fs · 2022-11-01

**Confidence:** 3
**Correctness:** 3
**Technical Novelty And Significance:** 3
**Empirical Novelty And Significance:** 3
**Recommendation:** 6

**Clarity, Quality, Novelty And Reproducibility:**

The specific approach proposed in this paper is novel to my knowledge, though I am not following offline RL literature very closely.

The writing is mostly clear, though it could benefit from improving the structure of the paper (see my comments on Section 3) and making the writing overall tighter.

The paper mentions, but does not elaborate on the details of the hyper-parameter search. This should be added to the Appendix.

Other suggestions to improve clarity:
- add calculations for Figure 1 in the Appendix (while easy to check on their own, it would help some in following the paper)
- I don't think p4 of Related work adds much to the paper, I would skip it and prioritize elaborating more on the methods section
- bold the best methods in Table 2, 3 and 4; maybe highlight which are sparse vs dense environments in Table 2
- clarify the meaning of the shaded area in Fig 4

**Strength And Weaknesses:**

The paper does a good job of defining and demonstrating the problems through both examples and the choice of environments. The experiments are fairly extensive and I appreciated the thorough discussion on the limitations of the method. The results are mostly consistent with the hypothesis that DT struggles with the stitching problem and CQL struggles with the delayed rewards.

Given the problems with DT are most acute when the training dataset does not contain (near) optimal trajectories, I found the experiment with an increased percentage of removed top trajectories (Fig 4) particularly interesting, and would appreciate seeing the same experiment run on multiple environments. It should be noted QDT starts outperforming DT only in the regime with 60-50% bottom trajectories. Overall QDT seems much closer to DT than CQL in performance. Why is that the case?

Since the main contribution is the method by which the returns are relabelled, Section 3 should be rewritten to be clearer and move the theory and pseudo-algorithm from the Appendix to Section 3. The lack of strong theoretical justification for the given procedure is one of the main weaknesses of the paper.

CQL outperforms DT/QDT on Maze2D even with sparse reward, which seems inconsistent with the initial hypothesis. Why is this the case? Separating dense and sparse environments in Table 2 would highlight this better.

I'd be also curious to see the results with more random seeds if the authors ran more experiments since the submission.

**Summary Of The Paper:**

This paper focuses on the problems with two popular offline RL algorithms: DT (which struggles with inferring optimal policies from partial solutions, referred to as the stitching problem) and CQL (which struggles to propagate rewards with a long time horizon). To address the shortcomings of both methods, the authors propose a method (QDT) that modifies returns in training trajectories with value functions (learned via CQL) when returns are lower than learned value functions, then uses this data to train DT.

On environments selected to test either stitching ability (Maze2D) or learning with sparse rewards (Mujoco with delayed reward), the authors demonstrate that, while QDT rarely outperforms both DT and CQL, it can learn when either of the two methods fails.

**Summary Of The Review:**

To improve the ability of DT to learn from partial solutions, the authors propose a method (QDT) that modifies returns in training trajectories with value functions (learned via CQL). On environments selected to test either stitching ability or learning with sparse rewards, the authors demonstrate that, while QDT rarely outperforms both DT and CQL, it can learn when either of the two methods fails.

The paper would be stronger if the proposed method had: (a) stronger theoretical justification, (b) better empirical results (regularly matching the best method in either setting), (c) the authors did a more thorough investigation into the performance difference between CQL and QDT (partly mentioned as left for future work in Section 6: conservative weight).

---

> ### Author Response · Authors · 2022-11-15
> **Reply to Reviewer J4fs (2)**
>
> > The lack of strong theoretical justification for the given procedure is one of the main weaknesses of the paper.
>
>  They main purpose of the paper is to empirically illustrate the potential of combined approaches as a potential way forward for the community, and although the choice of DT and CQL is not arbitrary, we see those as replaceable building blocks for our approach. However, we would like to note that when QDT employs DT and CQL, it inherits both their theoretical properties and limitations. Although the original DT paper also does not provide a theoretical justification for the approach, it reformulates RL task as a supervised learning task and the authors rely on the existing theoretical framework of statistical learning theory for general function approximators.
>
> > CQL outperforms DT/QDT on Maze2D even with sparse reward, which seems inconsistent with the initial hypothesis. Why is this the case?
>
> It is because maze2d involves simpler environments that have shorter episodes than the MuJoCo control tasks. Please refer to the following table:
>
> | Environment | Action Dim. | State Dim. | Ave. Good Episode Len. |
> | ----------- | :---------: | :--------: |----------------------- |
> | hopper      | 3           | 11         | 708.2 |
> | halfcheetah | 6           | 17         | 1000.0 |
> | walker2d    | 6           | 17         | 996.7 |
> ||||
> | maze2d-open | 2           | 4          | 49.8 |
> | maze2d-umanze | 2         | 4          | 128.6 |
> | maze2d-medium | 2         | 4          | 244.1 |
> | maze2d-large  | 2         | 4          | 314.6 |
> ||||
>
> This table shows that the action dimension, the state (observation) dimension and the episode length averaged over the top 5% returns in the dataset. It can be seen that MuJoCo tasks have higher action/state dimensions and longer episode lengths than Maze2d. Also, the evaluation results for the Sparse maze2d-medium and -large show some notable performance loss against the Dense counterparts, which is aligned with the fact that their episode lengths are longer than the maze2d-open and -umaze.
>
> We have added this content to Appendix E.1.
>
> > I'd be also curious to see the results with more random seeds if the authors ran more experiments since the submission.
>
> Following the suggestion, we have ran further experiments using different random seeds (five seeds in total) and updated Table2, 3 and 4 in the paper. The results are in line with the previous results in Table2, 3 and 4 with three random seeds.
>
> > The paper mentions, but does not elaborate on the details of the hyper-parameter search. This should be added to the Appendix.
>
> We use the same hyper-parameter values as the original papers (DT and CQL) as much as possible. However, we did hyper-parameter search for the CQL conservative weight. This is because the optimal conservative weight value could be different for CQL and QDT. We added the details and results in Appendix C.4.
>
> > * add calculations for Figure 1 in the Appendix (while easy to check on their own, it would help some in following the paper)
> > * I don't think p4 of Related work adds much to the paper, I would skip it and prioritize elaborating more on the methods section
> > * bold the best methods in Table 2, 3 and 4; maybe highlight which are sparse vs dense environments in Table 2
> > * clarify the meaning of the shaded area in Fig 4
>
> We have updated the paper to incorporate these suggestions. We hope the updated version correctly reflects these points.
>
> We agree that there is room for several improvements for this approach. Our paper provides a baseline and is the first work combining CQL and DT. Whilst there are alternative paths that could be followed to extend this work, we believe that the ICLR community (and our work) will greatly benefit from discussion at the conference.

---

> ### Author Response · Authors · 2022-11-15
> **Reply to Reviewer J4fs (1)**
>
> First of all, we would like to thank the reviewer for the comments and suggestions. We are pleased to receive such useful feedback, and we have updated our paper based on the advice. Please find our detailed response to the comments/questions below:
>
> > (Fig 4) particularly interesting, and would appreciate seeing the same experiment run on multiple environments.
>
> Following the suggestion, we have run the same experiment on the other two MuJoCo Gym environments. Please find the results (in plots) in Appendix E.4. Although they do not show a clear benefit of using QDT over DT, we argue this is due to the reason behind the gap between CQL and DT not just being the sub-optimality in the dataset (still the sub-optimality can be the reason behind the difference, but it is not the only cause in these cases). We also address this topic in Appendix E.3.
>
>
> > It should be noted QDT starts outperforming DT only in the regime with 60-50% bottom trajectories.
>
> We have changed the description in the paper to clarify that "QDT performs better than DT and becomes close to the CQL results at $X=40$ and $50$ (in the regime with 60-50% bottom trajectories)."
>
> > Overall QDT seems much closer to DT than CQL in performance. Why is that the case?
>
> The main reason is that QDT employs DT as its agent algorithm. The difference lays in its training data.
> If the environment/dataset has specific characteristics that work against DT approach, those also work against QDT. Some of these properties, such as dataset sub-optimality, are fixed/mitigated by QDT.
> However, there may be other elements that are against DT and QDT, e.g., the environment having a few critical states[1]. If this is also behind the gap between CQL and DT, then it is possible QDT performs close/the same as DT.
>
> Kumar et al. [1] studied the Dynamic Programming approach and the imitation learning approach and compared the upper bounds of their sub-optimality (the difference between the return
> from the optimal policy and the learned policy). They show that the Dynamic Programming approach is preferred over imitation learning when the environment has a few critical states --  the return of the episode mostly depends upon the actions in these states. The results in [1] are based on theoretical analysis (sub-optimality upper bounds). Hence, it is possible that the imitation learning approach (DT and QDT) can perform as well as or better than Dynamic Programming approaches (such as CQL) in practice. [2] empirically shows that the goal-conditioned approach remains competitive by selecting the right level of model capacity and the goal. There are still many open and ongoing discussions regarding the comparison.
>
> [1] Aviral Kumar, Joey Hong, Anikait Singh, and Sergey Levine. When should we prefer offline reinforcement learning over behavioral cloning? (2022)
>
> [2] Scott Emmons, Benjamin Eysenbach, Ilya Kostrikov, and Sergey Levine. Rvs: What is essential for
> offline rl via supervised learning? (2021)
>
> We added a brief discussion on this topic to the paper in Appendix E.3.
>
> > Since the main contribution is the method by which the returns are relabelled, Section 3 should be rewritten to be clearer and move the theory and pseudo-algorithm from the Appendix to Section 3.
>
> Thank you for the suggestion. We have moved the pseudo-algorithm and theory into Section 3. We have also re-worked some of its content to improve on clarity.

---

### Decision · Program_Chairs · 2023-01-20

**Decision:**

Reject

**Justification For Why Not Higher Score:**

Weak experimental results. Missing discussion around key works in offline RL, including those with the same motivation as this paper.

**Justification For Why Not Lower Score:**

N/A

**Metareview: Summary, Strengths And Weaknesses:**

The paper studies the offline RL setting. Here one key challenge for an agent is to learn to stitch near-optimal subtrajectories from the suboptimal offline dataset to constitute a near-optimal trajectory. It has been observed that return-conditioned BC models such as Decision Transformers struggle at stitching unlike dynamic programming based methods (eg, CQL). On the other hand, CQL struggles in environments with long horizons and sparse rewards. The proposed strategy in Q-learning DT (or QT) is sample: learn Q functions using CQL and use these as replacements for the return-conditioning in DT. Empirically, QDT oscillates somewhere between CQL and DT across the different range of environments.

The problem being addressed is also conceptually important and the reviewers uniformly appreciated the simplicity of the method and the clarity of exposition. The key shortcoming in the reviews that remains unaddressed is whether QDT is simply an interpolation strategy between DT and CQL, or a method that can achieve the best of both worlds. Empirical evidence suggests the former, where the performance of QDT is sandwiched between the two approaches. On a related note, I also found the paper lacking in discussion on offline RL methods more broadly and in particular, a relevant approach, TD3+BC [Fujimoto&Gu, NeurIPS 2021], is missing even a citation. This method is a standard baseline in most offline RL works and also aims to combine the complementary strengths of dynamic programming (TD3) and behavioral cloning (BC). In many empirical benchmarks (eg, https://arxiv.org/pdf/2112.10751.pdf), this baseline in fact outperforms DT and CQL, hence achieving the best of both worlds. While I appreciate the simplicity of QDT, I believe there is a deeper analysis vis-a-vis TD3+BC missing that could strengthen the paper discussion and perhaps inspire algorithmic improvements to QDT.